# Transits in Oncology: A Protocol Study for a Therapy-Educational Training Built-In Intervention

Carolina M. Scaglioso

Humanities Department, University for Foreigners of Siena, 53100 Siena, Italy; scagliosoc@unistrasi.it

**Abstract:** The study "Transits in oncology" has been perfected with the collaboration of the UOC of Oncological Mammary Surgery of the Azienda Ospedaliero Universitaria Senese Siena, specifically by Prof. Donato Casella. The study means to analyze the impact of art-therapy interventions aimed at minimizing psychological distress in women with a diagnosis of breast cancer/mammary carcinoma (anxiety/depression), hence improving their psychophysical wellbeing. To this end, the study employs the evaluation of specific psychological parameters with the purpose of monitoring anxiety and depression levels, while investigating a potential correlation between the anxiety and depression levels and other psychological variables, such as alexithymia. The mammary carcinoma diagnosis, to all effects, constitutes an actual "disorienting dilemma" for the woman: it leads to questioning one's way of life, and their past and future choices; the upheaval is conducive to a reflective phase that upsets one's "expectations of meaningfulness". The art-therapy intervention has been elaborated in a protocol that underscores its transformative methodology qualities: it aims to act on the regenerative potential of the turmoil, for an elaboration of trauma that does not negate it or further it (the feeling that nothing will change and everything will go back to the way it was before), but rather disrupts it. The final goal is to promote new existential practices, generating positive change towards self-awareness, stimulating the activation of one's latent resources by accessing one's symbolic world and one's imagination.

**Keywords:** health humanities; medical humanities; empathy in healthcare; narrative medicine; healthcare ethics

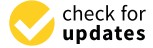



## 1. Introduction

The study "Transits in oncology" has been perfected with the collaboration of the UOC of Oncological Mammary Surgery of the Azienda Ospedaliero Universitaria Senese Siena, specifically by Prof. Donato Casella. The study means to analyze the impact of art-therapy interventions aimed at minimizing psychological distress in women with a diagnosis of breast cancer/mammary carcinoma (anxiety/depression), hence improving their psychophysical wellbeing. To this end, the study employs the evaluation of specific psychological parameters with the purpose of monitoring anxiety and depression levels, while investigating a potential correlation between the anxiety and depression levels and other psychological variables, such as alexithymia.

The mammary carcinoma diagnosis, to all effects, constitutes an actual "disorienting dilemma" for the woman: it leads to questioning one's way of life, and their past and future choices; the upheaval is conducive to a reflective phase that upsets one's "expectations of meaningfulness". The art-therapy intervention has been elaborated in a protocol that underscores its transformative methodology qualities: it aims to act on the regenerative potential of the turmoil, for an elaboration of trauma that does not *negate it* or *further it* (the feeling that nothing will change and everything will go back to the way it was before), but rather *disrupts it*. The final goal is to promote new existential practices, generating positive change towards self-awareness, stimulating the activation of one's latent resources by accessing one's symbolic world and one's imagination.

## 2. Study Description

### 2.1. Background and State of the Art

Breast cancer currently represents the cancer diagnosis with the highest incidence in women (30%), after skin cancers (Shi et al. 2014; Chen et al. 2016).

Numerous studies have shown that after diagnosis, women experience a more or less prolonged state of tension, so intense it may lead to experiences of anxiety, depression and fatigue (Samami et al. 2021; Schreier et al. 2019; Sharif 2017; Xiao et al. 2017; Zamanian et al. 2021). The onset of psychological symptoms in cancer patients is such a fairly common state that it can actually lead to consider depression and anxiety as neglected complications of the pathology (Lu et al. 2020). According to Pitman et al. (2018), existing data from cancer facilities indicate that the global prevalence of major depression, minor depression and anxiety, as defined by the Diagnostics and Statistical Manual of Psychiatric Disorders (DSM-5), is 14.9%, 19.2% and 10%. These figures far exceed those of the general population which are approximately 2% for major depression, 5% for minor depression and 7% for anxiety.

An additional peculiarity that has been found in subjects who have received a diagnosis of cancer is difficulty in identifying feelings and struggle in expressing emotions, which literature calls alexithymia (Marrazzo et al. 2016). Some authors consider alexithymia to be a reaction to the disease in newly diagnosed patients that declines differently in the various phases (Gritti et al. 2010; Ahrens and Deffner 1986). In this perspective, cancer patients' inhibited behavior can be considered as a time-limited reaction rather than a stable personality trait (Servaes et al. 1999). Therefore, alexithymia can be defined as a reaction to unpleasant emotional states, in which individuals limit their emotional range to mitigate painful experiences.

Several evidences suggest that alexithymia represents a risk factor in the onset of anxiety and depression (Saariaho et al. 2017) and, being a transdiagnostic factor (Lenzo et al. 2020), appears to be related to them. Furthermore, alexithymia has been proven to modulate psychological adaptation in women with breast cancer (Baudic et al. 2016) showing how positively correlated it is with maladaptive coping styles (Cho et al. 2020; De Gucht et al. 2004; Grassi et al. 2005; Servaes et al. 1999). Specific literature studies show how the anxiety and depression levels experienced are highly correlated to alexithymia, as well as to the representations that patients have of their disease (Jensen et al. 2014).

According to Grassi (2020, p. 1), "It is quite clear that cancer patients needs do not concern only the physical aspects related to the disease and its treatment, but a wide range of emotional, interpersonal and social implications, and that the consequences should be constantly monitored along the path of the disease". Therefore, it is essential to help cancer patients strengthen their ability to cope with their diagnosis and improve their emotional and psychological well-being, as demonstrated in the literature, since anxiety and depression significantly affect the quality of life of patients with breast cancer.

In this regard, the American Society of Clinical Oncology (ASCO) recently approved the guidelines of the Society of Integrative Oncology (SIO) for integrative therapies during and after breast cancer. These evidence-based guidelines recommend the use of integrative therapies for the management of symptoms and adverse effects, such as anxiety and stress (Elimimian et al. 2020).

Art-therapy is among one of the recommended therapies that have been shown to reduce anxiety and stress for breast cancer patients. Art-therapy is a wellness promotion intervention that employs the creative process of artistic invention to improve and enhance individuals' physical, mental and emotional well-being (Tang et al. 2019; Xu et al. 2020). Repercussions on the treated subjects' empowerment are clear, in terms of self-expansion and the development of one's latent resources, increasing perception of self-efficacy, motivation and tendency to develop an internal locus of control.

Literature records evidences supportive of art-therapy's use in different clinical and medical-psychological contexts (such as in Geue et al. 2010; Koom et al. 2016; Lefèvre et al. 2016; Puig et al. 2006; Tang et al. 2019) and in the pedagogical context in (Demetrio 1995,

2001; Goffi 2018; Guarino and Lancellotti 2017; Zimmerman 2000). Art-therapy's efficacy, particularly for a cancer patient, has been recognized in promoting emotional expression, consolidating interpersonal relationships and facilitating a reflection on the body image and, albeit in a not scientifically relevant way, in the reduction of anxiety and depression (Bosman et al. 2021; Ferrari et al. 2012). These results are also confirmed by studies which likewise highlight the importance of making the therapeutic process more human, through therapeutic and intervention tools capable of soliciting in the treated subjects the ability to cope with conflicts and adversities (Barbosa et al. 2007; D'Alencar et al. 2013; Elimimian et al. 2020; Elmescany 2010).

In our specific case, the study bases its development of writing's therapeutic use (Leedy 1973; Leedy and Rapp 1973; Mazza 2021). As for cancer patients, previous studies state that poetry therapy intervention may improve emotional resilience and anxiety levels (Tegnér et al. 2009). Group poetry therapy can improve moods and hopefulness in women with breast cancer and can have lasting positive effects in the long term (Daboui et al. 2018).

Writing in art-therapy can be used in different ways, chosen and adapted according to a person's characteristics and the therapeutic aims. In general, we can differentiate between an active and a passive mode. In the active mode, subjects are invited to compose poetic or literary passages, either freely or starting from a theme or key words indicated by the therapist. In this case, writing mainly has an expressive function and represents an important opportunity to get in touch with oneself and achieve greater self-awareness. The passive mode, on the other hand, requires the reading, according to a personal interpretation, of already existing passages. In this case, the function is mainly evocative, leveraging mechanisms of projection and identification. The use of writing is particularly suitable for people who face issues recognizing and expressing their emotions, such as alexithymic subjects and subjects with high levels of psychological distress. Numerous scientific evidences show how this technique brings a significant improvement in psycho-physical well-being (Merz et al. 2014; Solano 2001; Pennebaker 1997).

### 2.2. Aims

The study we present is in progress: we concluded for now the work phase with the retrospective group. As for the experimental group, we have reached up to this point one-third of the number of patients, with whom the three planned art-therapy sessions were concluded.

The research aims:

(1) To investigate the art-therapy intervention's impact in reducing psychological distress (anxiety/depression) of patients diagnosed with breast cancer and in promoting their psycho-physical well-being. Experienced anxiety and depression are the psychological parameters taken into consideration, monitoring their levels before surgery after the first and last art-therapy sessions;

(2) To investigate a possible relationship between anxiety and depression levels and other psychological variables, such as alexithymia.

We are achieving these aims through data analysis: the data derive from related psychological questionnaires that investiggoffiate the variables taken into consideration, both before and after the intervention of art-therapy. Patients' therapeutic paths are taken into consideration in the analysis of the variables, distinguishing in particular between those who only undergo surgery and those who are subjected to hormone therapy or chemotherapy in order to be able to discriminate their influence on the patient's emotional state. This data are compared with the data collected for the same questionnaires by a retrospective control group, for which their administration was deemed appropriate during the meetings with the psychotherapist.

### 2.3. Planning

The study is observational, non-profit, monocentric, retro-prospective, and lasted 24 months (of which 18 months were enrolment and 6 months were for data processing).

In the retrospective part, questionnaires completed by patients during the usual clinical practice in the period were taken into consideration.

Research Structure:

- Prospective part: during pre-hospitalization, the request for voluntary participation is expressed to patients who, due to the inclusion criteria provided by the study, could be included in the sample. Each patient is informed (Time 0, T0) through the information sheet about the purposes, objectives and methods with which the research will be carried out. Patients are offered to participate in three group meetings (with a maximum of ten participants per group) lasting two hours each on a weekly basis. The groups are led by an art-therapist and a psychologist-psychotherapist, online or face-to-face according to the health needs of the moment and to patients' needs. In detail: the first art-therapy meeting, defined as time 1 (T1), takes place after the third operating wound dressing, starting from the 21st day after surgery. The second art-therapy meeting, or time 2 (T2), will be set one week after the first. Finally, the third meeting, or time 3 (T3), will be held one week after T2, thus concluding the session of planned interventions. Patients will be subjected to questionnaires at different times: at T0, tests for assessing anxiety and depression (Hads), alexithymia levels (Tas-20) and, finally, the socio-anamnestic questionnaire will be administered. At T1, T2 and T3, after the art-therapy meeting, only the completion of the TAS-20 and HADS tests will be required.

These tests, based on clinical needs, are already widely used both pre-hospitalization and post-surgery in order to improve the specialist's therapeutic evaluation. For this reason, these tests correspond to the most suitable tools for the purpose of this study and will allow the comparison between the experimental group and the retrospective control group.

- Retrospective part: study of patients who filled out the same questionnaires, required by clinical practice, which were submitted to them at pre-hospitalization (T0) and on the 35th day post-surgery (T3).

To better define the art-therapy's role in the modification of the psychological variables considered, the data collected by the prospective group were compared with the data collected by the retrospective control group.

### 2.4. Variables

Depression and anxiety: measured by the Hads self-administered test (the Hospital Anxiety and Depression Scale) created specifically for measuring these variables in patients with organic diseases. The test consists of two scales of seven items each, one for the evaluation of anxiety, the other for depression. The time required for self-administration is about 10 min.

Alexithymia: measured by the Tas-20 self-administered test (Toronto Alexithymia Scale), consisting of 20 items (TAS-20). It is currently the most used tool for measuring alexithymia, as it is uniquely considered by researchers to be a reliable and valid measure. The expected time for self-administration is about 10 min.

### 2.5. Tools

The *20-Toronto Alexithymia Scale*, Italian version TAS-20 (Bressi et al. 1996), is a 20-item self-assessment scale based on a 5-point Likert scale, which measures the construct of alexithymia. Three dimensions of the TAS-20 describe the main components of alexithymia: "struggle in identifying emotions", "struggle in describing emotions" and "outward-oriented thinking". An overall score identifies the degree of alexithymia. Alexithymic people are defined as those who obtain a score higher than or equal to 61; borderline those who have a score between 51 and 60; and non-alexithymic subjects who obtain a score lower than or equal to 50.

*Hospital Anxiety and Depression Scale* (Zigmond and Snaith 1983). The tool was developed with the aim of providing researchers and clinicians with a practical, reliable and valid

tool to identify situations of psychological distress, in particular anxiety and depression. It is a self-administered questionnaire composed of 14 items; specifically, seven items make up the Anxiety scale, and the remaining seven make up the Depression scale. The response mode is on a four-level Likert scale (range 0–3), and the final scores on the two subscales can vary from 0 to 21, with a nine-point cut-off indicating that the symptoms of anxiety and depression are sufficient to justify a specialist intervention. The instructions for filling in the questionnaire require you to refer to how the subject has felt over the last week. The items that make up the Depression subscale do not include somatic indicators of Major Depressive Disorder according to the DSM IV, as they may be manifestations of organic disease or drug treatment.

*The Socio-anamnestic questionnaire* is administered for the detection of socio-demographic factors and social factors, built according to the directives of the experimental centre (UOC of the University Hospital of Siena, Breast-Unit) and can be completed in about two minutes.

### 2.6. Study Population

The study involves women diagnosed with breast cancer belonging to the Breast Unit of the Sienese University Hospital. The inclusion and exclusion criteria for the prospective group are the same as those for the retrospective group:

Inclusion criteria:

- Age between 18 and 65 years;
- Female patients diagnosed with breast cancer;
- Level of education not lower than eighth grade;
- Native language proficiency (if patients are not Italian native speakers, a level of linguistic competence equal to or higher than level C1 is required);

Exclusion criteria:

- Have undergone neo-adjuvant therapy;
- Presence of cognitive disorders of any nature;
- Presence of psychiatric pathologies that cause severe impairment in the social and work spheres (disorder in the schizophrenia spectrum and other psychotic disorders; bipolar disorder of the first and second type).

To avoid the bias typical of experimental research, random sampling will be carried out in order to reach the same number of patients for both groups.

### 2.7. Sampling Size

Power calculation was carried out with a G-power data analysis software. The power analysis was conducted taking into consideration a minimum effect size (0.1) $\alpha = 0.05$, $p = 0.80$ in order to detect even the smallest effects of the effectiveness of art-therapy in the levels of anxiety and depression. This analysis made it possible to establish that the total number of subjects to be recruited appropriate to obtain significant differences is 138 (69 experimental group, 69 retrospective control group).

### 2.8. Data Management and Statistic Plan

Data are initially collected in paper form through questionnaires collection and then transferred to electronic support. Data are set in a pseudo-anonymous form, or through the attribution of an identification code that does not make them directly attributable to the interested parties. The collected data are then fed to electronic spreadsheets (Excel) and to a specific statistical program (SPSS) in order to process them for the purposes envisaged by the study.

Description and processing of data collected for each of the variables' measurement scales will be carried out through the use of descriptive statistics. This process will make it possible to identify, both for the control and for the experimental group, the average means with the relative standard deviations or the medians with their interquartile ranges. In

addition to the specific subdivision of the two study groups, subgroups will be created based on the different therapy to which the patients will be subjected in order to exclude any influence on the outcome of the results. Based on the distribution of the data, parametric tests such as the t-Student or non-parametric tests such as the Mann–Whitney will be used to highlight the differences between groups. In particular, once the test has been selected, the differences will be determined, both between the control group of the experiments and within each individual group and between the values collected at time 0 and those collected at the end of each art-therapy meeting. Finally, the linearity relationship between the psychological variables measured through the Pearson correlation index will be studied. Statistical significance will be considered for *p*-values below 0.05.

## 3. Protocol

Meetings will be held under the guidance of two psychotherapists: a conductor and a co-conductor with the role of observer.

The protocol can be divided into two specific moments that are part of the same process. The first moment is more individual, and the second involves working in a group.

At the beginning of each meeting, lasting two hours, patients receive four highly evocative images that illustrate four significant myths of paths of awareness and rebirth. The selection of these myths is the result of research and brainstorming with the psychotherapists team and the art-therapist who will conduct the meetings with the patients. Texts were therefore produced by the same team: they decided to stress brevity and the words' evocative capacity. Psychotherapists decided to not worry about any simplifications since the privileged aspect is not the specific understanding but the resonance the myths may arouse. The images were selected from paintings with a figurative, not abstract, style, with recognizable figures and with widely described landscapes and backgrounds. The myths are as follows: in the first meeting, Deucalion and Pyrrha, Thyramus and Pisbee, Proserpine, and Aracne and Minerva; in the second meeting, Perseus, Cadmus, Mercury and Argus, and the Myrmidons; in the third meeting, Filomene and Bauci, Europe, Cadmus and Harmony, and Semele and Bacchus. In fact, the close connection between images and mindset is known: the mind learns through image-processing, and through them it orients itself in the emotions and nourishes the conscience. Therefore, not only "every psychic process is an image and an imaginative act, without which no consciousness could exist", but "if the world does not take the form of a psychic image, it is practically non-existent" (Hillman [1992] 2002, p. 24).

In a second moment, we hand over the written text corresponding to the chosen myth. 2A. Women are invited to read it quickly, without dwelling on contents, and to highlight words that resonate in their minds; other words may be erased as they like (by drawing on them, making scribbles, lines, etc.). 2B. On a different sheet, each woman writes the words that she was attracted to, and then (2C.) she chooses three of the selected words. 2D. Anyone who wants can share both the 'resonant' words for her and the three words finally selected. 2E. Each woman writes each of the three words in three different sheets, which she then folds and places inside a green container provided by the psychotherapist. 2F. The women are invited to let their imagination run wild, to build a story together starting from the incipit "Once upon a time . . . " offered by the psychologist: the story will be fictional, like a fairy tale, and it will be the story of the group for that encounter (the creative use of language sets in motion processes based on abductive thought, which authorizes multiple descriptions of objects, events, and sequences, urging the search for unthinkable solutions: (Bateson 1979)). One woman is then invited to draw a word from the container and begin with it; followed by all the others who continue the story in the same way, until all words are exhausted. The observer writes the story, which is then read again and opened to the group for comments: What kind of impression does it make on you? Would anyone change it and at what point and how? Do you feel it is yours? What kind of emotions did you feel? How did you feel?

Only at the end of the third and final meeting, as a synthesis and greeting (2G.), all the sheets with the words used are taken and placed inside a basin of blue coloured water, a symbol of flow, letting go and rebirth.

Three meetings are certainly not enough to constitute an area of intervention in which transformative learning can generate new existential practices. The long-term perspective, however, to be calibrated according to the results that will be obtained in this study underscores the chance of using art-therapy in a stable and structured way as a transformative methodology that urges us to go through physical consistency to overcome it. We enhance the images' and words' generative value and their excess functions (pushing beyond the organization of the existing cognition to create another form and other cognitions), which are neither predictable nor preordained.

What is achieved in each meeting (the story) starts from a textual (iconic and verbal) solicitation that gives shape to one's feelings; the interactions that follow allow for an emergence that is the result of concrete interactions, of intertwined rhythms, of different realities that meet in the continuous alternation of points of contact and points of 'breaking'. Different trajectories of each one flex and incorporate the trajectories of the others in a generative key, to create a text that does not follow predetermined routes, passes through the unknown and allows you to lightly access the 'extraordinary' of your own poetic space.

### 3.1. The Group: Reasons Why

The meeting is both a container and a privileged place for observation.

Group modality, if well led, amplifies individual resources and favours all the reciprocal aspects such as recognition, solidarity, sharing and respect for the plurality of points of view in addition to the production of meaning.

Leadership skills are essential to contain emotion, maintain interest and advance the dynamic flow of interactions. In this way, the group can obtain results that are more difficult to achieve individually, for example, a deep emotional and cognitive understanding thanks to the interaction between the different experiences on an imaginative level. In the group, moreover, it is easier to question the elements of personal rigidity in order to consider possible alternatives; the well-organized group can become a wiser and vigorous supra-individual and collective mind than each one of the individual components, capable of building new meanings through the synergy generated by the encounter.

### 3.2. Tales: Reasons Why

A tale, like a disease, and above all like the oncological disease, has its own time and space. Tale-time is a suspended one, characterized by indefinite, very long durations. Illness-time is no longer the same as that which came 'before': it is framed by uncertainty, being a suspended time as well. Uncertainty concerns healing times, recovery times and whether there will actually be a recovery and if it will be complete. Uncertainty is also caused by the fact that the patient finds herself having to wait for the results of examinations and checks, that others decide for her and take care of her: the patient is in a grip of anxiety caused by a diagnosis that is still dangerous today; she only asks to be reassured that she will be able to resume her 'old life' as soon as possible and archive the disease as an episode to be forgotten. Illness-time is suspended in waiting and in the hope that the previous life can be restored.

Secondly, a tale, such as an oncological disease, is qualified by a space which is also uncertain, indefinite: distances, dangers to face, goals to reach, symbolic places to gain in the tale are in the oncological disease the hall of histological samples, the room with pre-surgery records, the operating room, the ward, etc.

At the end of the tale, an authentic metaphor of training tells of a path of construction both for the subject and the world, often using the narrative *topos* of the journey, or of an initiatory path that is in effect a path of self-recognition. Along the journey, the main character must face up to their responsibilities, endure the test of detachment and the fear of loneliness and death. Similarly, in her illness, the patient experiences that she is alone

in the world: no one can replace her in suffering and in the sense of abandonment, in fear of not becoming well and of death. As in a tale, the oncological disease evokes from the deep structures of the unconscious phantoms of anxieties and expectations, which take on a strong symbolic, cognitive and emotional significance.

Weaving tales together helps to face the indefinite, to know and recognize fear, to understand it not only as a constitutive element of one's person but of the human being. As it happens to the main characters of the stories, every upheaval of order, every abandonment of certainty may lead to worlds with a wideness not even remotely glimpsed until one was stuck in the fear of transforming oneself, of changing. No story is about stillness and immobility, but rather deals with travels and transformations, research and discoveries: such is, in health and in sickness, the peculiarity of the human being.

## 4. From the Observational Study, the Perspective of an Integrated Intervention Program

After the turning point of Alma Ata in 1978, the concept of health increasingly took on procedural and relational connotations of the well-being concept: it concerns all aspects of human existence, both the intra-personal and the inter-personal dimensions.

Well-being, as an overall and interactional phenomenon, is measured both on clinical and biological data that refer to the physical conditions of the subjects and on the subjective and emotional perception they have of their state of health; therefore, how much the perception of well-being can change according to the life stages of a subject or when special events occur must be taken into account.

According to these premises, the wellness paradigm's assumptions bring attention to educational work that can allow subjects to draw on their potential to acquire and/or increase empowerment strategies and implement feelings and positive energies, even in the condition of illness. Treatment procedures in turn no longer belong to just the assistance field but are considered real communicative events, aimed at healing patients, yet also at increasing their personal awareness and autonomy. As a consequence of this, medical practice has progressively become aware of the need to take a relational approach that enhances the meaning of the experience and the sense of the vital worlds of the subjects being treated, their identity, their role, their way of communicating, etc.

Speaking of educational intervention and well-being in the cancer patient may seem farfetched, even today, despite the advancements of therapies, survival and remission rates. The present study's assumptions are, however, those expressed by Dewey, according to which any experience is educational that, by opening up persons to new experiences, is capable of improving them (Dewey 1938). Furthermore, the literature has been enriched with multiple and authoritative voices on the regenerative potential of upheaval (among all, Freud 1919; Moroni 2019; Morelli 2020). The pain, the 'damage', is at the origin of human consciousness. They find no sense if not elaborated through the justifications of religions: suffering, misdirection and upheaval annihilate thought.

Trauma, however, can also be a revealing element if the subject is capable of cultivating an establishing thinking mode that allows them to go through internal and relational conflicts and embrace change, abandoning habits oriented towards denial or waiting for everything to go back as it was before. Both denial and expectation are passive, de-responsible positions and favour inertia/rejection strategies, typical of an unintended subject who feels they do not have to and cannot be a protagonist. On the other hand, an ardent and unwavering intentionality can lead to stiffness and disavowal of one's self and of events, and it does not recognize space for ungovernability and complexity. The educational task that was never before so necessary in health and 'normality' conditions is to support and help the subjects to remain in instability, learn to let go, always remain aware and in contact with oneself and be available to change without, however, abandoning the search for one's own meaning.

Breast cancer diagnosis (and the following path) is a peremptory "disorienting dilemma" for women, to which none of her experiences and knowledge are able to offer a solution.

Practices of care and training may try to transform pain and "humanize it" (Demetrio 2001), making it less devastating. The competent intervention of care workers (medical, psychological, educational) is capable of tracing the extreme fragility of the sick woman to the 'acceptable' fragility that characterizes us as all human beings and make them live through the dilemma without being annihilated and without becoming lost.

This study believes that it is possible and useful to gently accompany women who, each in their own way, suddenly find themselves experiencing a phase of reflection that often involves their "meaning perspectives" and that critically evaluates the content, process and premises of their efforts to interpret an experience, understand it and give it meaning (Mezirow 2000). The diagnosis can, in fact, lead to an authentic learning, with new awareness and change, that questions the assumptions at the basis of personal reference frames that are no longer suitable for interpreting new experiences. Hence, the perspective we glimpse at the end of the study presented, which remains observational in nature, is the structuring of a subsequent path in which the woman is supported in this critical re-examination of her thoughts and her intentionality through art-therapy. The patient is also assisted in the disruption of the "border structures" of her system of meaning, which are no longer capable of anchoring, through routine habits of expectations, new experiences to old consolidated meanings (Mezirow 2000). The path is therefore located on several levels, both psychological and educational: a real training path, which has the intention of triggering a cognitive experience with resonances on all levels of being.

The recognition and strengthening of the empowerment skills in patients is a primary goal of the recommended path. In fact, the cancer patient may recognize it as important to care for herself through nutrition or to improve her lifestyle. Yet, she is in any case forced by her illness to 'hand over' the restoration of her full health to other people. Doctors are in charge of choosing the path to follow, from the first blood sample to the type of surgery, to supportive and adjuvant therapies if needed. The exclusion from the real choices that concern her can elicit in the patient a sense of abdication that makes her resign from being an active, re-active and pro-active person aware of her own abilities.

A psychological intervention with artistic mediation can instead be useful to support increases in self-esteem and in the sense of self-efficacy and self-determination: in short, an awareness of the process of their individual potential evolution in which resources can be recovered that are often difficult to call on in moments of crisis but still remain latent in everyone, with results that can sometimes exceed expectations.

A care system offering the integration of a path such as the one we described, on the other hand, aims to promote the recognition that care is always characterized by educational reciprocity, by reciprocal role investiture and by 'interdependence'. On one hand, the patient recognizes her need for help by conferring power to the carer with a heroic act typical of adulthood, that is, the acceptance of limits (Demetrio 2001); on the other, the caregiver takes responsibility for the patient, respectfully recognizes the dimension of 'Person' and becomes aware that every action taken retroacts and remodels them too, shaping a generative relationship of reciprocal co-evolution and transformation.

The educational path that we see as a consequential continuation of the observational study takes into account phases of transformative learning identified by the aforementioned Mezirow (self-analysis, self-assessment, comparison with others, exploration of new options, building trust, building new skills, attempts and finally integration of the perspectives of meaning). Art-therapy interventions, in fact, do not help to work through adjustments of meaning schemes, nor through their innovation (which does not produce effective changes in points of view), but precisely on the transformation of the perspectives of meaning, that is, on the refocusing of their system. Art-therapy refocuses the perspectives system's meaning, which is necessary to activate the reflection through which we reach awareness of the specific assumptions on which distorted, partial and no longer adaptive perspectives of meaning are based; it then proceeds in their transformation through a reorganization of meaning (Mezirow 2000).

The operator role is essential in this particular process: in today's complex world, characterized by an increased need for transits and transitions, the adult of the 'normal' population already has issues ruling their own life processes and training processes and their exercise of self-reflection, freedom and a critical spirit. More than ever, the same can be said for a woman who has become a cancer patient, who often focuses on this traumatic and painful event without being able to grasp its transformative potential and learning opportunities. And it cannot be otherwise. Criticism (Newman 2014) that was placed on the transformative theory, specifically on the effective neutrality and impartiality of the operator-trainer ("active agents of cultural change": Mezirow 2000, p. 30) accompanying the work, refers to the debate on the neutrality of education, which we do not dwell on because it goes beyond the theme of the contribution and the space we have at our disposal: we limit ourselves to reiterating that no educational intervention is neutral and that educating requires recognizing that education is ideological and remembering, among the extensive literature on the subject, the thinking of Jerome Bruner (Bruner 1996) and Paulo Freire (Freire 1996) that education, however much one may try to argue the contrary, is never neutral, but always political in a broad sense, full of social and economic consequences.

It is therefore essential that the operator not only have a composite professionalism expressed in solid psycho-pedagogical and socio-health skills, but that they (1) are able to give without reservations and in all conditions the consideration and respect that is due to each patient; that they (2) are consistent and 'connected', available to the relationship even when defensive attitudes arise, and are 'controlled' but capable of communicating in an authentic and transparent manner; and that they (3) are capable of cognitive empathy, that is, to participate deeply in the experience of the other, avoiding processes of identification and projection.

The integration of interventions, such as the ones outlined in care paths, can improve the well-being of patients, and the quantitative results of this study can justify this. However, it is also important to underline how much the system perspective can change through these paths: the oncological health facility frees itself from the image of a provider of rigidly prepared protocols and services and becomes an animator, a place where the treatment intervention responds to the deepest needs and to the expectations of the subjects, recognizing the person's multifactorial and plural dimensions. Treatment does not respond only with more properly medical, surgical and rehabilitative protocols to questions raised by cancer patients and to their lack of meaning, but also with protocols rich in symbolic resources, in helping relationships that welcome and recognize for the term "Cure" both the meaning of concern/attention and the semantic reference to anxiety, pain and anguish. These are precisely the aspects that the study aims to highlight, those aspects that the patient offers as an expression of her psychic suffering and also of her search for meaning (Grignoli 2008): aspects that must receive as much attention as that given to the physical ones.

**Funding:** This research received no external funding.

**Institutional Review Board Statement:** Not applicable.

**Informed Consent Statement:** Not applicable.

**Data Availability Statement:** Not applicable.

**Conflicts of Interest:** The author declares no conflict of interest.

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
