# Peer review of "Transits in Oncology: A Protocol Study for a Therapy-Educational Training Built-In Intervention"

_humanities, doi:10.3390/h11060136_

Round 1

Reviewer 1 Report

The conceptual framework of this study is very compelling and the authors do a good job of substantiating their theoretical principles with existing literature. This paper appears to be working with subjects in Italy. This reviewer is from the USA where the description of their methodology could also be included as a creative arts therapy method know as bibliotherapy or poetry therapy. This reviewer is making the suggestion that they include some research from this field as well. There are some sentence structure concerns also making this reviewer wonder if English is their second language. Eg page 9 line 426 ... empowerment skill is not a secondary goal..." It is not clear why the word NOT is used. Also the switching of pronouns from she to him to she again is confusing, in this long sentence. This is a very interesting and exciting paper and this reviewer supports its publicatoin. 

Author Response

First of all Thank You very much for Your encouragement, suggestions, help and opinions: they have been extremely relevant for reviewing my work, and sparked new ideas on its continuation.

I included several researches useful to better frame the protocol within the bibliotherapy/poetry therapy context (line 107). I also made the long phrase at p.9 (line 426, now 429) more understandable, and only with female nouns.

Reviewer 2 Report

This is an excellent article which describes an interesting and original interdisciplinary study that will be of considerable interest to readers.  Focussing on women with breast cancer The study addresses the impact of art therapy interventions and the distress experienced by women with breast cancer. The study protocol is clearly outlined, detailed  and convincing. Its dual purpose of monitoring anxiety and depression levels, while investigating a potential correlation between the anxiety and depression levels and other psychological variables, such as alexithymia, addesses an important and neglected area of womens mental health. The paper is very well-written and engaging.  It will be of interest to a wide readership including art therapists; psychotherapists, practitioners involved in cancer care  and their patients. I  look forward to reading the results.

There is a typo on page 268 "mith' should be myth.

I have some reservation about the use of the word 'training' however this is addressed on pages 347-348. as follows:

'In the end the tale, an authentic metaphor of training, tells a path of construction both for the subject and the world, often using the narrative topos of the journey, or of an initiatory path that is in effect a path of self-recognition'.

Author Response

First of all Thank You very much for Your encouragement, suggestions, help and opinions: they have been extremely relevant for reviewing my work, and sparked new ideas on its continuation.

I corrected “myth” at line 268 (now 271). As the reviewer noted, the choice of the “education” term is challenged and explained at 347-348 line, therefore I left the term unchanged.